# A Prospective Study of the Quality of Life of Patients with Stress Incontinence before and after a Transobturator Tape (TOT) Procedure—Preliminary Report

**DOI:** 10.3390/jcm10194571

**Published:** 2021-10-01

**Authors:** Maciej Zalewski, Gabriela Kołodyńska, Anna Mucha, Waldemar Andrzejewski

**Affiliations:** 1Department of Gynaecology and Obstetrics, Faculty of Health Sciences, Medical University of Wrocław, 50-367 Wrocław, Poland; zalewskim@interia.pl; 2Independent Public Health Care Center of the Ministry of the Interior and Administration in Wroclaw, Department of Gynaecology, 50-233 Wrocław, Poland; 3Department of Physiotherapy, Wroclaw University of Health and Sport Sciences, 51-612 Wrocław, Poland; waldemar.andrzejewski@awf.wroc.pl; 4Faculty of Medical Sciences and Health Sciences, University of Social and Medical Sciences in Warsaw, 04-367 Warszawa, Poland; 5Department of Genetics, Wrocław University of Environmental and Life Sciences, 50-375 Wrocław, Poland; anna.mucha@upwr.edu.pl

**Keywords:** stress urinary incontinence, quality of life, TOT procedure

## Abstract

Background: Urinary incontinence (UI) is a significant social problem. The latest figures show that it affects as many as 17–60% of the female population, and it is one of the most common chronic diseases. Incontinence substantially decreases the quality of patients’ lives. The transobturator tape (TOT) procedure is the gold standard in surgical treatment due to its high efficacy and low complication rate. Objective: The aim of this study was to assess the quality of life (QoL) of patients with stress incontinence before and after the TOT procedure. Method: The study included 57 patients diagnosed with stress incontinence on the basis of ultrasonography and history. The QoL before and after surgery was measured using the Incontinence Impact Questionnaire (IIQ-7) and the Incontinence Quality of Life (I-QOL) standardised questionnaires. Results: The IIQ-7 scores for each question were higher (indicating poorer quality of life) before surgery than after surgery. The results for almost all domains were statistically significant. The I-QOL results also showed that, in most cases, the quality of patients’ lives improved after the surgery. Statistically significant changes were observed in all three questionnaire domains of avoidance/limiting behaviour, psychosocial impact, and social embarrassment. Conclusion: Surgical treatment of stress incontinence with TOT results in resolution of bothersome symptoms in the majority of patients, leading to improved comfort in life.

## 1. Introduction

Urinary incontinence is a serious social problem. The exact number of patients affected by this condition is unknown, because there are different definitions of this disorder and the classification systems are not uniform. As the latest data show, urinary incontinence affects as many as 17–60% of the female population, making it one of the most common chronic diseases [1]. Estimates suggest that the number of patients is even higher, but that patients do not report their complaints due to embarrassment and a common belief that incontinence is a natural condition associated with ageing. The average incidence of this condition is 27.6% in women, and 10.5% in men [2]. The most common type is stress urinary incontinence (SUI), which constitutes as many as 50% of all incontinence cases among women [3].

Urinary incontinence is defined as involuntary leakage of urine through the urethra due to dysfunction of the closure mechanism of the bladder. In stress incontinence, this is typically induced by a sudden increase in the intra-abdominal pressure (IAP) [4]. Other symptoms include enuresis (i.e., urination without the urge), no change in the frequency of micturition compared with the period before the disease, and resolution of symptoms during sleep. The etiopathogenesis of this condition has been attributed to factors that impair the transfer of intra-abdominal pressure to the junction between the urethra and the bladder, and to the proximal urethra [5].

Factors contributing to stress urinary incontinence include the following [6]:age;vaginal delivery;disturbed innervation of the pelvic floor;disturbed structure of connective tissue;hypoestrogenism;obesity;chronic pulmonary diseases.

Although women’s awareness of this condition is increasing year by year, many patients still consider it a natural symptom of ageing.

Urine loss significantly reduces the quality of life in SUI patients and affects both their physical and mental well-being [7]. Due to the accompanying symptoms, people with urinary incontinence are very often forced to limit their social roles and activities. They often withdraw from social and family life due to bothersome symptoms. A major problem is also that patients are often ashamed to talk about their condition with others, for example, their family or medical staff. Many women also have the mistaken belief that urinary incontinence is a natural phenomenon which is closely related to pregnancy, childbirth and the ageing process. The impact of incontinence on quality of life largely depends on the severity of the symptoms.

With acute symptoms, basic life activities are impaired. In his research, Hunskaar estimated that depressive symptoms occur in as many as 80% of respondents with severe incontinence, whereas in patients with mild symptoms that number oscillates around 40% [8]. Irwin et al. showed that patients who suffer from incontinence very often experience a sense of loss of control over their bodies, which further increases their psychological discomfort and reduces their life activity [9].

In a different study, Milsom assessed the impact of SUI on performing daily tasks. Nearly 100% of the patients indicated that this problem reduced their quality of life. Over 50% of the respondents said that these ailments hindered their daily functioning. As many as 70% of the women surveyed said that they could not travel longer than 30 min because of this condition. The researcher observed the occurrence of emotional disorders in 45% of the patients. Moreover, he pointed out that urinary incontinence also has a direct impact on the deterioration of the quality of life of their families [10].

The problem of urinary incontinence also affects the economy. In a study carried out in Great Britan, Turner et al. investigated the costs associated with urinary disorders. They qualified 23,183 people for the study. The findings showed that the total annual expenditure incurred by the National Health Service was 233 million pounds, while individual patient costs were estimated to be 178 million pounds. The study demonstrates that urinary incontinence has a measurable impact not only on the national budget, but also on every patient. The financial expenses involved in the purchase of sanitary napkins, nappies and pant diapers are very high and constitute a major problem for patients with urinary incontinence. Many women who feel more comfortable when they have additional protection for their underwear or clothing use them preventively [11].

The latest reports indicate that latent urinary incontinence is also extremely common. It is estimated to affect 36–80% of patients with pelvic floor prolapse and is associated with urethral flexion or pressure, potentially leading to micturition disorders [12]. These cause closure of the urethral lumen, and consequently a lack of immediate compression of the bladder, which can trigger urine leakage through the urethra. Consequently, in this group of patients, the the symptoms of urinary incontinence in this group of patients thus only occur only after surgical reconstruction of the pelvic floor. Currently, reports on simultaneous surgery for pelvic organ prolapse and urinary incontinence yield contradictory results. Performing these two procedures at the same time has been linked to a higher number of complications [13]. There is no consensus on the management of this group of patients. It is, certain, however, that the extent of the operation should be decided on with the patient after history and physical examination, and after they have been informed of all therapeutic options. Improving the quality of the patient’s life should be a priority [14].

One of the most frequently used techniques in the surgical treatment of SUI is a mid-urethral sling via the retropubic or transobturator approach. Randomised controlled studies showed a similar success rate between TVT (80.8%) and TOT (77.7%) [15,16]. The transobturator tape procedure (TOT) via the transobturator route is frequently performed due to low rates of complications, including bladder or bowel injury. Huang et al. compared the effectiveness of TOT and TVT in women with SUI. They found that TOT may have better outcomes than TVT in terms of operation time and hospital stay. TOT showed fewer complications and less blood than TVT, but there was no significant difference between the two methods. There was a greater decrease in IIQ-7 and UDI-6 scores in the TOT group than in the TVT group. This indicated that the quality of life of patients after the TOT procedure was better [17].

The definition of quality of life was proposed by the World Health Organization in 1994 states that the QoL is an individual’s perception of their position in life with regard to the culture and the value system in which they live. Factors that directly affect the QoL include somatic health, interpersonal contacts, and the community characteristics that are essential for a particular person [18]. It is significant that physicians today focus not only on extending patients’ lives, but also on improving their quality of life. Modern medicine aspires to restore the patient’s QoL from before the disease. Hence, there is an increasing interest in the assessing of the QoL of people affected by various diseases [19].

At present, not only strictly medical goals, but also non-medical goals are crucial elements of the therapeutic process. The latter include improving the patient’s well-being and functioning, both physically and socially. QoL studies are part of the holistic approach [20]. In medicine, both subjective and objective assessment of the QoL should be performed. Subjective assessment of the QoL can be based on questions addressed to patients. Nevertheless, this assessment depends on patients’ mental state, preferences, system of values, personality traits, etc. Objective assessment of QoL, on the other hand, is usually performed using standardised questionnaires. These are valuable, repeatable instruments that measure patients’ QoL and the effectiveness of treatment [21].

In the available literature, there are not many reports that assess the quality of life of patients 12 months after the TOT procedure. In this study, the authors sought to assess the long-term impact of the procedure on women’s quality of life. In most cases, researchers evaluate the effectiveness or correctness of TOT in terms of quality of life a few weeks after the surgery. As TOT is one of the most frequently used methods in the treatment of SUI, there should be many reliable studies assessing their impact on the quality of life also in the long-term perspective.

The aim of this study was to assess QoL in patients with stress incontinence before and after stress incontinence surgery using the TOT method.

## 2. Material and Methods

### 2.1. Design and Data Collection

Eighty patients were qualified for the study, but due to the long duration of the research, both sets of questionnaires were finally obtained from 57 patients who had been diagnosed with stress urinary incontinence on the basis of ultrasound examination and history at the gynaecology ward at the Hospital of the Ministry of Internal Affairs and Administration in Wrocław. Urodynamic examination was performed in patients with doubtful diagnosis. The patients were qualified for surgery based on literature guidelines. Additionally, all patients gave their informed written consent to take part in the study prior to entering the project. All the participants completed the IIQ-7 and I-QoL questionnaires twice: once upon admission to the ward and 12 months after the surgery. The patients were qualified for the study on the basis of the inclusion and exclusion criteria.

The inclusion criteria were:stage II and stage III stress incontinence confirmed by ultrasound examination and an interview;not taking hormone replacement therapy (HRT) before or after the surgical procedure;informed written consent of the patient to take part in the project.

The exclusion criteria were:women with overactive bladders (OAB) or mixed urinary incontinence (MUI);women with urinary tract fistulas;women with congenital or acquired defects of the urethra or the bladder;women with urinary tract infections;women taking medication that contributes to an overactive bladder.

All eligible patients underwent the TOT suburethral sling procedure, which decreases urethral hypermobility and consequently eliminates or reduces the number of involuntary urine leakage episodes caused by physical effort. The TOT procedure, proposed by Delorme in 2001, is considered the gold standard in the surgical treatment of stress incontinence, due to its high effectiveness and low complication rate. We used this method because it is as effective as the tension-free vaginal tape approach but takes half the time [22]. The management principles employed during the surgery included the following:insertion of a bladder catheter;incision and dissection of the vaginal mucosa and fascia;proper placement of the tape;preventing the implants from curling or rolling up;avoiding implant infections;optimal tension-free suturing of the vaginal walls [23].

### 2.2. Measures

All the patients enrolled in the project were given the IIQ-7 and IQOL questionnaires. The questionnaires were filled in on admission to the ward, before the TOT procedure and 12 months after the procedure at the follow-up examination. The number of responses to particular questions varied, as some patients marked two or none of the responses that they felt best described their quality of life at that time.

The Incontinence Impact Questionnaire (IIQ-7) is used to assess the quality of life of women with urinary incontinence. The IIQ-7 questionnaire is an abridged version of the IIQ questionnaire, which contains 30 questions. All seven questions in the IIQ-7 questionnaire concern the impact of urinary incontinence on patients’ lives. The questions refer to the ability to do household activities, physical activities, entertainment, travel, leisure, and mental health. Each question was rated on a scale of 0 to 3, where 0 = not at all, 1 = a little, 2 = medium, 3 = very. The average of the scores obtained is calculated [24]. Only those questions that were answered are taken into account. The average, which ranges from 0 to 3, is multiplied by 33 1/3 to convert the results to a scale of 0 to 100. A higher result indicates worse symptoms and poorer quality of life [25].The Incontinence Quality of Life (I-QOL) questionnaire is a commonly used validated QoL instrument specific for urinary incontinence. It consists of 22 questions divided into three domains: avoidance/limiting behavior, psychosocial impact, and social embarrassment. Each item can be rated on a five-point scale where 1 = extremely, 2 = quite a bit, 3 = moderately, 4 = a little, and 5 = not at all [26]. The total I-QOL score and the three domain scores are calculated by summing the unweighted item scores and transforming them into a 100-point scale, where 0 = no problem, and 100 = most severe [27].

## 3. Statistical Analysis

Statistical analysis was performed using R Project software [28]. The statistical significance of the impact of surgery on the patient’s QoL was tested. The results of the IIQ-7 and I-QOL questionnaires, which the patients were asked to fill in prior to the surgery and twelve months after the surgery, were used for this purpose. Categorical variables, illustrating the number of individual patient responses to each question, were compared using Fisher’s exact test. The statistical significance of the differences in the domain scores (total, avoidance and limiting behaviors, psychosocial impacts, social embarrassment) within each group (before and after the surgery) was tested using the Wilcoxon test for paired samples. The statistical tests were carried out with a significance level of *p* = 0.05. A non-parametric test was used due to the incompatibility of the distribution of the analysed variables (domain scores) with the normal distribution, which was verified with the Shapiro–Wilk test at the significance level of *p* = 0.05.

## 4. Results

The project involved 57 women with a diagnosis of stage II and stage III stress incontinence. The mean age of the patients was 70.28 years, and the range was 61–87 years.

The number of responses to individual questions differed, which resulted from the specificity of the questionnaires used. In the IIQ-7 questionnaire, when the symptoms did not affect the activity indicated in the question, the patient did not mark the answer to the question that would explain its impact. On the other hand, in the IQOL questionnaire, the range of responses to each question ranged from 0–5.

The analysis of the results obtained from the IIQ-7 questionnaire shows that the operation did not significantly affect the improvement of the quality of life of the respondents (Table 1).

The answers obtained in questions 1–7 show that the surgery also increased the patients’ ability to function in everyday life. The obtained results were not statistically significant (Table 2).

In most cases, the I-QOL results also confirmed that the patients’ QoL improved after the surgery. Statistically significant changes were observed in the three questionnaire domains, namely, avoidance/limiting behavior, psychosocial impact, and social embarrassment (Table 3). The majority of statistically significant changes were found in the avoidance/limiting behavior domain (Table 4). A statistically significant change was also observed in the psychosocial impact and social embarrassment domains, although not in all the questions (Table 5 and Table 6). This may be due to the fact that, even prior to surgery, the patients claimed that QoL in these domains was not as important for them as in the avoidance/limiting behavior domain.

## 5. Discussion

The main purpose of this study was to assess the quality of patients’ lives before and after TOT surgery for stress urinary incontinence. The findings demonstrated that the patients’ QoL substantially improved after the surgery, as the symptoms that had substantially limited everyday functioning subsided in most cases. In their study, Ellerkmann et al. measured the QoL of patients with anterior bladder wall prolapse and urinary incontinence. Both their findings and ours clearly show that urogynaecological problems have a great impact on the quality of patients’ lives [29]. Digesu et al. demonstrated considerable differences in the symptoms associated with pelvic organ prolapse between women with urinary incontinence and those without this health problem. Their study also revealed a weak correlation between vaginal anterior wall prolapse and urinary system symptoms, except for the ‘feeling of incomplete emptying of the bladder’ and ‘effort during urination’. This may have been caused by the fact that the bladder outlet is obstructed by the uterus, bladder, or rectum prolapsing throughvaginal skin [30].

As reported by Nilsson et al. and Lee, urinary incontinence noticeably lowers women’s QoL, irrespective of age, and has a negative impact on women’s relationships [31,32]. According to Riss and Kargl, stress incontinence has a great impact on the QoL and should also be considered when starting treatment [33]. Bushnella et al. provided evidence that urinary incontinence is not only a simple physiological problem, but also an issue that should be considered in the context of social, physical, and emotional consequences [34]. Some studies show that women with urinary incontinence restrict their physical and social activity for fear of uncontrolled leakage. Furthermore, women with incontinence are significantly more likely to report sexual problems, which, however, can be significantly improved with effective treatment [35]. Our study confirms this by showing improvement in the QoL in this aspect after surgery.

The effect of urinary incontinence on mental health was examined by Goldacre et al., who demonstredted that women with urinary incontinence are more prone to depression and self-harm. Women with advanced symptoms of urinary incontinence suffered from severe depression three times as often as those with mild incontinence. Moreover, the relationship between urinary incontinence and anxiety symptoms is also higher in this group of patients. This problem has a significant impact on women’s social life, especially in the case of older patients. Women with pelvic organ prolapse feel humiliated and embarrassed, which frequently leads to withdrawal from family and social life. Some researchers have emphasized that depression and urinary incontinence might be strongly correlated with each other [36]. Oh and Ku concluded that there is a strong correlation between urinary incontinence and mental disorders. Although urinary incontinence is not directly linked tomortality, it causes anxiety, depression, and dissatisfaction with life, which can lead to suicide attempts. The prevalence of depression in this group of patients is similar to that observed in patients with diabetes, cardiac conditions, and other chronic diseases [37].

Both this study and those cited above show how important it is to assess the QoL in patients with stress incontinence. Successful treatment not only improves patients’ physical well-being by eliminating an anatomical defect, but also positively affects their mental health. The results presented here show that healthcare professionals should not only provide surgical treatment but also assess and promote patients’ QoL and mental health. It therefore seems necessary to carry out further research in order to develop strategies for the prevention, early diagnosis, and treatment of patients with stress incontinence.

## 6. Conclusions

Surgical treatment of stress incontinence with the TOT method may alleviate bothersome symptoms and improve quality of life.The IIQ-7 and the I-QOL questionnaires are useful instruments that measure the effectiveness of treatment and help patients understand how their QoL improved after surgery.Further research seems necessary to develop strategies of managing patients who suffer from stress incontinence and have lower QoL.

## 7. Limitations

Our paper has some limitations. Our findings may rely on an insufficient number of subjects and therefore this research requires a follow-up. However, our limitation was obtaining a homogeneous group of women who met the study inclusion criteria. To our knowledge, this is the first paper to analyse the relationships between the TOT procedure and the quality of life in women before surgery and 12 months after surgery.

## Figures and Tables

**Table 1 jcm-10-04571-t001:** Domain scores of the IIQ-7 questionnaire from women before and after surgery.

	Before Surgery	After Surgery	*p*-Value
**Total Score**	46.1538 (0–100) 48.1605 (24.1045)	46.1538 (0–96.1538) 44.5221 (26.0864)	0.5303

Data are presented as median (range) and mean (standard deviation). Given *p*-value is for the *t*-test.

**Table 2 jcm-10-04571-t002:** Responses from IIQ-7 questionnaire from women before and after surgery.

Question	Response	Before Surgery	After Surgery	*p*-Value
How much do the ailments affect your ability to do household chores (cooking, housecleaning, laundry)?	not at all	1	1	0.2862
	slightly	11	8	
	moderately	10	10	
	greatly	12	3	
How much do the ailments affect your physical recreation such as walking, swimming, or other exercise?	not at all	0	1	0.2314
	slightly	12	4	
	moderately	12	12	
	greatly	15	9	
How much do the ailments affect your entertaining activities (movies, concerts, etc.)?	not at all	1	1	0.7183
	slightly	10	5	
	moderately	11	7	
	greatly	6	7	
How much do the ailments affect your ability to travel by car or bus more than 30 min from home?	not at all	1	0	0.6077
	slightly	12	5	
	moderately	7	7	
	greatly	10	8	
Do the ailments affect your participation in social activities outside your home?	not at all	14	6	0.9366
	slightly	7	3	
	moderately	6	4	
	greatly	1	0	
How much do the ailments affect your emotional health (nervousness, depression, etc.)?	not at all	1	0	0.8061
	slightly	12	6	
	moderately	11	10	
	greatly	11	8	
How much do the ailments affect your feelings of frustration?	not at all	2	1	0.2042
	slightly	14	3	
	moderately	10	9	
	greatly	12	10	

Data are presented as subgroups size. Given *p*-values are for the Fisher’s exact test.

**Table 3 jcm-10-04571-t003:** Domain scores of the IQoL questionnaire from women before and after surgery.

	Before Surgery *n* = 57	After Surgery *n* = 57	*p*-Value
Avoidance and limiting behaviours	35 (0–80) 36.71 (25.72)	22.50 (0–80) 27.92 (25.05)	0.02015
Psychosocial impacts	32.22 (2.22–82.22) 35.04 (25.80)	18.89 (0–73.33) 24.48 (21.58)	0.002981
Social embarrassment	36 (0–80) 38.13 (25.55)	28 (0–80) 30.07 (27.19)	0.01584

Data are presented as median (range) and mean (standard deviation). Given *p*-values are for the Wilcoxon test for dependent samples.

**Table 4 jcm-10-04571-t004:** Avoidance and limiting behavior responses from IQoL questionnaire from women before and after surgery.

Question	Response	Before Surgery	After Surgery	*p*-Value
I’m worried that I won’t be able to get to the toilet on time	1	18	25	0.03326
	2	7	11	
	3	10	10	
	4	13	2	
	5	12	12	
I am afraid of coughing/sneezing due to urinary incontinence	1	15	22	0.1561
	2	4	10	
	3	12	10	
	4	11	6	
	5	18	12	
I need to control myself when I’m getting up from a sitting position	1	27	39	0.05056
	2	7	7	
	3	12	9	
	4	9	1	
	5	5	4	
I worry about where the toilets are in a new place	1	18	26	0.1898
	2	11	14	
	3	10	3	
	4	4	4	
	5	17	13	
It is important for me to be able to use the toilet frequently	1	14	25	0.04097
	2	5	5	
	3	10	6	
	4	17	6	
	5	14	18	
It is important for me to plan every detail in advance because of urinary incontinence	1	20	22	0.3786
	2	10	7	
	3	10	7	
	4	5	12	
	5	15	12	
I have difficulty in resting properly at night because of urinary incontinence	1	25	30	0.7271
	2	8	10	
	3	8	7	
	4	9	5	
	5	10	8	
I have to watch how much fluid I drink due to urinary incontinence	1	17	30	0.08735
	2	8	8	
	3	13	6	
	4	5	6	
	5	17	10	

Data are presented as subgroups size. Given *p-*values are for the Fisher’s exact test.

**Table 5 jcm-10-04571-t005:** Psychosocial impacts responses from IQoL questionnaire from women before and after surgery.

Question	Response	Before Surgery	After Surgery	*p*-Value
I feel depressed because of urinary incontinence	1	18	31	0.04457
	2	11	4	
	3	8	9	
	4	8	9	
	5	15	7	
I don’t feel comfortable enough to be out of the house for a long time because of urinary incontinence	1	14	29	0.005231
	2	11	2	
	3	8	11	
	4	16	13	
	5	11	5	
I feel frustrated because urinary incontinence restricts me from doing what I like	1	18	31	0.1281
	2	12	8	
	3	12	11	
	4	8	6	
	5	10	4	
Urinary incontinence is “still in my head”	1	19	25	0.7467
	2	7	8	
	3	11	9	
	4	7	7	
	5	16	11	
Urinary incontinence makes me feel sick	1	21	35	0.08521
	2	14	6	
	3	7	6	
	4	8	4	
	5	10	9	
I feel helpless because of urinary incontinence	1	20	29	0.182
	2	13	10	
	3	7	11	
	4	7	3	
	5	13	7	
Urinary incontinence reduces the feeling of joy of life	1	22	27	0.09636
	2	7	14	
	3	10	4	
	4	11	11	
	5	10	4	
Urinary incontinence limits my possibilities of choosing clothes	1	29	37	0.6324
	2	7	5	
	3	9	7	
	4	6	3	
	5	9	8	
I’m afraid of having intercourse because of urinary incontinence	1	32	35	0.1654
	2	8	4	
	3	6	1	
	4	5	1	
	5	9	6	

Data are presented as subgroup size. Given *p*-values are for the Fisher’s exact test.

**Table 6 jcm-10-04571-t006:** Social embarrassment responses from IQoL questionnaire from women before and after surgery.

Question	Response	Before Surgery	After Surgery	*p*-Value
I’m afraid that people around me smell urine	1	23	40	0.0001028
	2	9	2	
	3	10	0	
	4	3	6	
	5	15	10	
I’m afraid that this problem will increase with age	1	14	23	0.07386
	2	9	8	
	3	7	1	
	4	7	11	
	5	23	17	
I’m afraid of being humiliated because of urinary incontinence	1	19	38	0.00163
	2	15	3	
	3	8	4	
	4	6	3	
	5	12	11	
I’m afraid of uncontrolled urine leakage	1	12	22	0.05753
	2	11	3	
	3	11	6	
	4	11	13	
	5	15	16	
I feel I can’t control my bladder	1	14	23	0.3665
	2	13	7	
	3	11	10	
	4	8	6	
	5	14	14	

Data are presented as subgroups size. Given *p*-values are for the Fisher’s exact test.

## Data Availability

The datasets used and/or analyzed during the current study are available from the corresponding author on reasonable request.

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
