# Peer review of "A Prospective Study of the Quality of Life of Patients with Stress Incontinence before and after a Transobturator Tape (TOT) Procedure—Preliminary Report"

_jcm, 2021, doi:10.3390/jcm10194571_

Round 1
Reviewer 1 Report
In this report, authors investigated and reported the quality of life (QOL) of patients with stress urinary incontinence before and after the transobturator tape (TOT) procedure using Incontinence Impact Questtionnaire (IIQ-7) and the Incontinence Quality of life (I-QOL). The conclusions are not new, as it has already been reported that the improvement of urinary incontinence by TOT also improves QOL. In this study, the number of cases was only 57, and the questionnaires did not seem to be sufficiently collected. The description of the methods was inadequate, and there seemed to be problems with the statistical analysis and interpretation of the results.
・On page 4, line 18-23, authors mentioned “the majority of patients have decreased the frustration felt earlier due to urinary incontinence symptoms (question 7). In addition, after surgery, patients began to undertake physical and recreational activities more often (question 2). The answers obtained in question 1 show that the performed operation also significantly increased the patients’ ability to perform home tasks (question 2)”. According to Table 2, There were no questions that were significantly different in p-value, and the authors' description does not seem to be correct.
・On page 4, line 14, authors mentioned “The project involved 57 women”. In Table 1, the total number of responses before surgery was considerably less than 57, and the total number varied from 28 to 44. Furthermore, the total number of responses was much lower after surgery, and the total number varied from 13 to 26. There is no explanation in the text for this variation in the number of responses, and the statistical analysis was conducted without any description of the handling of missing values. Therefore, we cannot judge whether the results are correct or not.
・Despite the fact that Table 3 lists n=57, the total number of responses in Tables 4, 5, and 6 was generally 60, with some places having 47, 58, and 59 cases. There was no explanation in the text for this discrepancy in the Tables.
Author Response
Dear Reviewer,
Enclosed herein I am Submitting the revised manuscr ipt Ref. No .: jcm-1383326 . The text has been changed according to reviewers ' suggestions . Changes in the manuscript are marked .
Thank you very much for your thorough evaluation of my manuscript , valuable comments and help in improving .
With kind regards ,
Gabriela Kołodyńska
Reviewer # 1 :
In this report, authors investigated and reported the quality of life (QOL) of patients with stress urinary incontinence before and after the transobturator tape (TOT) procedure using Incontinence Impact Questtionnaire (IIQ-7) and the Incontinence Quality of life (I-QOL ). The conclusions are not new, as it has already been reported that the improvement of urinary incontinence by TOT also improves QOL. In this study, the number of cases was only 57, and the questionnaires did not seem to be sufficiently collected. The description of the methods was inadequate, and there seemed to be problems with the statistical analysis and interpretation of the results.
Answer :
The introduction explains why the presented article differs from those on similar topics:
“ There are not many scientific reports in the available literature that would assess the quality of life of patients 12 months after the TOT pro- cedure . In this study , the authors wanted to assess the long -term impact of the procedure on the quality of life of women . In most cases , researchers evaluate the effectiveness of the procedure in terms of its correctness or quality of life a few weeks after the procedure . Due to the fact that the TOT procedure is one of the most frequently used methods in the treatment of SUI, there should be many reliable studies assessing their impact on the quality of life also in the long -term perspective .
The aim of this study was to assess QoL in patient "
The Limitations explains why the number of subjects was so small:
Our paper has certain limitations . The results obtained by the authors may be related to the insufficient number of women in the studies and therefore they need to be continued . However , a certain limitation was to obtain a homogeneous group of postmenopausal women who meet the study inclusion criteria . To our knowledge , this is the first paper analyzing the relationships between TOT procedure and quality of life in postmenopausal women before and 12 months after surgery
In addition, the title added that this is a preliminary report and the research is still continued.
The description of the methods was inadequate , and there seemed to be problems with the statistical analysis an d interpretation of the results - have been corrected
Reviewer # 1:
On page 4, line 18-23, authors mentioned “the majority of patients have decreased the frustration felt earlier due to urinary incontinence symptoms ( question 7). In addition , after surgery , patients began to undertake physical and recreational activities more often ( question 2). The answers obtained in question 1 show that the performed operation also significantly increased the patients ' ability to perform home tasks ( question 2) ”. According to Table 2, There were no questions that were significantly different in p-value, and the authors ' description does not seem to be correct.
Answer :
Although no statistically significant differences were noted , correct parameters were noticed , therefore the authors of the manuscript wanted to describe these changes. Nevertheless, agreeing with the reviewer's suggestion, they were removed.
Reviewer # 1:
On page 4, line 14, authors mentioned “The project involved 57 women ”. In Table 1, the total number of responses before surgery was considerably less than 57, and the total number varied from 28 to 44. Furthermore , the total number of responses was much lower after surgery , and the total number varied from 13 to 26. There is no explanation in the text for this variation in the number of responses , and the statistical analysis was conducted without any description of the handling of missing values. Therefore , we cannot judge whether the results are correct or not.
Despite the fact that Table 3 lists n = 57, the total number of responses in Tables 4, 5, and 6 was generally 60, with some places having 47, 58, and 59 cases . There was no explanation in the text for this discrepancy in the Tables .
Answer :
The question of the difference in responses following the reviewer's suggestion is explained in the Material and methods subchapter
All patients qualified to the project completed the IIQ-7 and IQOL questionnaires on their own. Questionnaires were filled out on admission to the ward, before the TOT procedure and 12 months after the procedure during the follow-up examination. The number of responses to particular questions varies, as some patients marked two or none of the responses that, in their opinion, best described their quality of life at a given time.

Reviewer 2 Report
Thanks for the opportunity to review A prospective study of the quality of life of patients with stress incontinence before and after a transobturator tape (TOT) procedure.
Introduction
- Can you make the general discussion more stress urinary incontinence SUI focused as opposed to UI globally please? For example SUI effects younger women post-partum so is very relevant to the social impact (women in the work force etc.). Any data on financial impact to society of SUI? Can also discuss the proposed success of TOT in the treatment of SUI (60-70% satisfaction); therefore reason to believe it has a large impact on QOL.
Materials and methods
- How does one use U/S to diagnose SUI? What is the data to back this method up? Why is it not part of AUA guidelines in work-up of SUI? Please explain.
Tables
- Capitalize “Before Surgery” etc.
Discussion
- According to the European guidelines TOT is no longer standard of care for SUI- TVT is. Could you please include a paragraph on how you think the results from this study will transfer to those outcomes of the TVT as well?
- Any limitations to the study?
- Any studies to look at impact of QOL after POP +/- TVT/TOT repair? Is it even bigger?
Author Response
Dear Reviewer,
Enclosed herein I am Submitting the revised manuscr ipt Ref. No .: jcm-1383326 . The text has been changed according to reviewers ' suggestions . Changes in the manuscript are marked .
Thank you very much for your thorough evaluation of my manuscript , valuable comments and help in improving .
With kind regards ,
Gabriela Kołodyńska
Reviewer # 2 :
Introduction
Can you make the general discussion more stress urinary incontinence SUI focused as opposed to UI globally please ? For example SUI effects younger women post- partum so is very relevant to the social impact ( women in the work force etc.). Any data on financial impact to society of SUI? Can also discuss the proposed success of TOT in the treatment of SUI (60-70% satisfaction ); therefore reason to believe it has a large impact on QOL.
Answer :
The introduction has been changed as recommended by the reviewer. We expanded it with the indicated information. Thank you very much for your suggestions.
Reviewer # 2:
Materials and methods
How does one use U / S to diagnose SUI? What is the data to back this method up ? Why is it not part of AUA guidelines in work-up of SUI? Please explain .
Answer
We haven t use it to diagnose SUI but in our opinion it s good method to confirm it .
References :
Huang WC, Yang JM (2003) Bladder neck funneling on ultrasound cystourethrography in primary stress urinary incontinence: a sign associated with urethral hypermobility and intrinsic sphincter deficiency . Urol 61 (5): 936–941
Tunn R, Goldammer K, Gauruder-Burmester A, Wildt B, Beyersdorff D (2005) Pathogenesis of urethral funneling in women with stress urinary incontinence assessed by introital ultrasound . Ultrasound Obstet Gynecol 26 (3): 287–292
Masata J, Martan A, Halaska M, Otcenasek M, Voigt R (1999) Ultrasound imaging of urethral funneling . Int Urogynecol J Pelvic Floor Dysfunct 10 (S1): S62
Dietz HP, Clarke B (1998) The urethral pressure profile and ultrasound parameters of bladder neck mobility . Neurourol Urodyn 17 (4): 374–375
Dietz HP, McKnoulty L, Clarke B (1999) Translabial color Doppler for imaging in urogynecology : a preliminary report. Ultrasound Obstet Gynecol 14: 144–147
Dietz HP, Clarke B (2001) Translabial color Doppler urodynamics . Int Urogynecol J Pelvic Floor Dysfunct 12 (5): 304–307
Reviewer # 2:
Tables
Capitalize " Before Surgery " etc.
Answer
We changed it as recommended by the reviewer. Thanks for your suggestion.
Reviewer # 2:
Discussion
According to the European guidelines TOT is no longer standard of care for SUI- TVT is . Could you please include a paragraph on how you think the results from this study will transfer to those outcomes of the TVT as well?
Any limitations to the study ?
Any studies to look at impact of QOL after POP +/- TVT / TOT repair ? Is it even bigger ?
Answer
According to European Association of Urology and IUGA Mid-urethral synthetic sling inserted by either the transobturator or retropubic route provides equivalent patient- reported outcome at five years , so in our article we also focused on TOT. We added information about this in the introduction .
We have added a subsection on limitations . Thank you for pointing out the lack.
In the presented article, we wanted to consider only the TOT method. In the future, we would like to write an article assessing QOL after POP +/- TVT / TOT .

Round 2
Reviewer 1 Report
In this report, authors investigated and reported the quality of life (QOL) of patients with stress urinary incontinence before and after the transobturator tape (TOT) procedure using Incontinence Impact Questtionnaire (IIQ-7) and the Incontinence Quality of life (I-QOL). The conclusions are not new, as it has already been reported that the improvement of urinary incontinence by TOT also improves QOL. In this study, the number of cases was only 57, and the questionnaires did not seem to be sufficiently collected. The description of the methods was inadequate, and there seemed to be problems with the statistical analysis and interpretation of the results.
・In the method section, authors mentioned that “Patients were quantified based on guidelines from the literature, which include:•urogynecological history; •urinalysis; •physical examination; •cough trial performed in a lithotomy position; •evaluation of urethral mobility; •evaluation of urine retention after micturition; •transurethral ultrasound; •urodynamic examination in doubtful cases”; however, the results were not shown in result section.
・On page 4, line 14, authors mentioned “The project involved 57 women”. In Table 1, the total number of responses before surgery was considerably less than 57, and the total number varied from 28 to 44. Furthermore, the total number of responses was much lower after surgery, and the total number varied from 13 to 26. T the statistical analysis was conducted without any description of the handling of missing values. Therefore, we cannot judge whether the results are correct or not.
・Authors mentioned in the conclusion section that “1.Surgical treatment of stress incontinence by the TOT method relieves the majority of patients of bothersome symptoms and improves their QoL. 2. The IIQ-7 and the I-QOL questionnaires are useful instruments that measure the effectiveness of treatment and help patients understand how their QoL improved after surgery.” According to Table 2, the results of IIQ-7 did not show the significant improvement after TOT; therefore, the conclusion No.1 would be overstated.
・In the Limitations section, authors mentioned that “this is the first paper analysing the relationships between TOT procedure and quality of life in postmenopausal women before and 12 months after surgery”; however, neither the methods nor the results of this paper state that the study was conducted only in postmenopausal women.
Author Response
Dear Reviewer,
Enclosed herein I am Submitting the revised manuscr ipt Ref. No .: jcm-1383326 . The text has been changed according to reviewers ' suggestions . Changes in the manuscript are marked .
Thank you very much for your thorough evaluation of my manuscript , valuable comments and help in improving .
With kind regards ,
Gabriela Kołodyńska
Reviewer # 1 :
Comments and Suggestions for Authors
In this report, authors investigated and reported the quality of life (QOL) of patients with stress urinary incontinence before and after the transobturator tape (TOT) procedure using Incontinence Impact Questtionnaire (IIQ-7) and the Incontinence Quality of life (I-QOL). The conclusions are not new, as it has already been reported that the improvement of urinary incontinence by TOT also improves QOL. In this study, the number of cases was only 57, and the questionnaires did not seem to be sufficiently collected. The description of the methods was inadequate, and there seemed to be problems with the statistical analysis and interpretation of the results.
Answer:
Dear reviewer, thank you very much for all your comments and suggestions. They contributed significantly to the improvement of the quality of our manuscript. We made changes according to the detailed directions you described below.
Reviewer # 1 :
・In the method section, authors mentioned that “Patients were quantified based on guidelines from the literature, which include:•urogynecological history; •urinalysis; •physical examination; •cough trial performed in a lithotomy position; •evaluation of urethral mobility; •evaluation of urine retention after micturition; •transurethral ultrasound; •urodynamic examination in doubtful cases”; however, the results were not shown in result section.
Answer:
As indicated by the reviewer, we have removed this snippet. Because he could actually mislead the reader. The specification of these features was intended only to show which criteria of qualification for the TOT procedure are applied in accordance with the literature guidelines. We agree with the reviewer that they may have been considered by the reader as grouping factors or even as inclusion criteria for the studies that are described in the following paragraphs.
The aim of our research was not to assess the effects of the operation in relation to the criteria qualifying for the TOT procedure, but to assess the impact of the operation on the quality of life of patients who underwent it.
Reviewer # 1 :
・On page 4, line 14, authors mentioned “The project involved 57 women”. In Table 1, the total number of responses before surgery was considerably less than 57, and the total number varied from 28 to 44. Furthermore, the total number of responses was much lower after surgery, and the total number varied from 13 to 26. T the statistical analysis was conducted without any description of the handling of missing values. Therefore, we cannot judge whether the results are correct or not.
Answer:
The differences in the number of responses mentioned by the reviewer result from the specificity of the questionnaires used. On the advice of the reviewer, we have added a description in the Results section before the tables that explains the discrepancies to the readers:
„The number of responses to individual questions differed because it resulted from the specificity of the questionnaires used. In the IIQ-7 questionnaire, when the symptoms did not affect the activity indicated in the question, the patient did not mark the answer to the question that would explain its impact. On the other hand, in the IQOL questionnaire, the range of responses to each question ranged from 0-5.”
In the statistical analysis section, we have supplemented the description of the methods used to make them clearer.
Reviewer # 1 :
・Authors mentioned in the conclusion section that “1.Surgical treatment of stress incontinence by the TOT method relieves the majority of patients of bothersome symptoms and improves their QoL. 2. The IIQ-7 and the I-QOL questionnaires are useful instruments that measure the effectiveness of treatment and help patients understand how their QoL improved after surgery.” According to Table 2, the results of IIQ-7 did not show the significant improvement after TOT; therefore, the conclusion No.1 would be overstated.
Answer:
Thank you very much for this suggestion. Indeed, the conclusion was exaggerated. We improved it:
„Surgical treatment of stress incontinence by the TOT method may alleviate bothersome symptoms and improve quality of life.”
Reviewer # 1 :
・In the Limitations section, authors mentioned that “this is the first paper analysing the relationships between TOT procedure and quality of life in postmenopausal women before and 12 months after surgery”; however, neither the methods nor the results of this paper state that the study was conducted only in postmenopausal women.
Answer:
Thank you very much for this suggestion. Indeed, the indication of women only in the postmenopausal period was included here by mistake.
Reviewer 2 Report
None
Author Response
Dear Reviewer,
thank you very much for your valuable comments and suggestions that have made our manuscript better.
Kind regards,
Authors
Round 3
Reviewer 1 Report
Although this paper states that it is a prospective study, it does not describe a sufficient research plan. There is no description of the background of the collected cases, nor is there a sufficient accumulation of cases, and therefore, satisfactory statistical analysis cannot be performed. From the beginning I thought this paper was not of sufficient quality to deserve publication in this journal, but in past reviews I have written constructive comments for future use. However, if you don't fundamentally redo the data collection for your paper and prepare the paper's style properly, the quality of your paper will not improve even if you fix the surface expressions.